# Long-Term Assessment of Captive Chimpanzees: Influence of Social Group Composition, Seasonality and Biographic Background

**DOI:** 10.3390/ani13030424

**Published:** 2023-01-26

**Authors:** Arnau Pascual, Elfriede Kalcher-Sommersguter, David Riba, Dietmar Crailsheim

**Affiliations:** 1Research Department, Fundació MONA, 17457 Girona, Spain; 2Fundació UdG: Innovació i Formació, Universitat de Girona, 17003 Girona, Spain; 3Institute of Biology, University of Graz, 8010 Graz, Austria; 4Facultat de Lletres, Universitat de Girona, 17003 Girona, Spain; 5Departament de Psicologia, Facultat d’Educació i Psicologia, Universitat de Girona, 17004 Girona, Spain

**Keywords:** chimpanzee, social network analysis, behavior, group alterations, captive care, group composition, seasonality, adverse early life experiences

## Abstract

**Simple Summary:**

Wild chimpanzees flexibly adapt their behavior based on many social and environmental aspects of their lives. These include seasonality and food availability as well as aspects regarding their communities and parties, such as group size, sex ratio and the presence of sexually receptive females. This results in a stimulating but also stressful life. On the contrary, housing conditions in captivity are far less stimulating, often lacking the enriching social component of fluctuating group compositions found in wild fission–fusion societies. Providing captive chimpanzees with an adequate social environment is crucial to ensure their wellbeing. We had the rare opportunity to analyze behavioral data of an all-male chimpanzee group that experienced two major group alterations, i.e., the integration of two adult females and the passing away of two adult males. Our findings highlight the importance of conducting longitudinal observations to objectively record variations in the chimpanzees’ behavior. Furthermore, we were able to demonstrate the impact of their social environment, i.e., group size and sex ratio, on the chimpanzees’ activity levels and occurrence of abnormal and self-directed behaviors.

**Abstract:**

Wild chimpanzees (*Pan troglodytes*) live in flexible fission–fusion societies with frequent changes in both group size and composition. These changes depend mostly on resource availability and individual social preferences yet in captivity are determined by housing organizations. During a period of seven years, we observed a group of sanctuary chimpanzees, focusing on how they adapted to changes in the group composition over time. Using linear mixed models (LMMs), factors such as group size, sex ratio, seasonality, and the individuals’ sex and origin (wild caught vs. captive born) were considered in order to evaluate the impact on the chimpanzees’ activity levels, the occurrence of undesired behaviors (abnormal and self-directed behaviors) and the social grooming networks. Our results indicate that the activity levels and the occurrence of undesired behaviors were impacted by changes in group composition and the individuals’ biographic background. The colder season was marked by higher levels of activity and more time spent grooming. Moreover, it was the individuals’ origin but not changes in group composition that affected social grooming, with wild-caught chimpanzees grooming far less frequently. Long-term observations are essential to evaluate, predict and detect potential benefits and/or issues of housing conditions while considering the social and physical environment.

## 1. Introduction

Primates, especially chimpanzees (*Pan troglodytes*), are among the most exhibited and popular animals in zoos and animal collections as well as protagonists in entertainment events and TV commercials [1]. What makes these animals especially interesting in the eye of the public are their similarities to humans [2], their cognitive capacities [3] and their social complexity [4,5]. However, said aspects also implicate difficulties for housing organizations in terms of providing species-adequate environments and care in captivity, while striving for high levels of wellbeing. While the definition, evaluation and detection of wellbeing issues in captivity have advanced greatly over the past few decades [6], moving from concepts such as the Five Freedoms and Five Domains to a Life Worth Living [7], there still are many aspects that need to be addressed, both by the scientific community as well as by professionals in captive animal care. In particular, the idea of offering primates more “control and choice” [8,9] regarding their lives in captivity appears especially interesting yet difficult to achieve. While housing organizations may make an effort in providing a physical environment and care management procedures restoring some degree of control and choice for these animals, two of the most limiting aspect of captivity remain, i.e., constraints of the enclosure’s size/complexity and social group composition. While the control and choice of said aspects in free-ranging chimpanzees are influenced by ecological and environmental factors, such as the presence of neighboring chimpanzee communities [10,11], resource availability [12], the presence/absence of predators and partially human presence in their natural habitat [13], in captivity these are mostly controlled by the housing organization. Evidently, the physical and social environment directly impacts the chimpanzee’s wellbeing and potential to exhibit species-typical behaviors.

Chimpanzees are highly social animals, being one of the most socially complex species among all non-human primates [14]. They are structured in societies [15] consisting of multi-male multi-female communities, ranging in size between 20 and 150 individuals. Upon reaching sexual maturity, females migrate from one community to another while males remain in their natal group [14,16,17,18,19,20]. They have a fission–fusion social organization [21,22,23] in which communities break off into flexible, smaller interchangeable subgroups [14,24,25]. Such subgroups may reunite with other subgroups in order to fuse or shuffle individuals on a daily basis but might also remain apart for prolonged periods of time. These frequent transfers of individuals between small subgroups, allow the efficient adaptation of the group size and composition according to resource availability [14,26,27,28] in order to increase the survival rate and group functionality [27,29]. The size of these subgroups may vary depending on social and ecological factors, such as quantity and quality of food resources [14,22,30], predation pressure [31] or the presence of sexually receptive females, and the need to compete over available resources [24,32,33,34,35]. Such flexible group divisions are an effective mechanism to reduce intra-group competition, i.e., frequent fights and wounding [36], and provide greater flexibility in the exploitation of food resources [37,38]. Furthermore, they allow females to avoid aggressive males and males to monitor females in estrus [37]. Lehman and Boesch [27] demonstrated that demographic variables can also have a strong impact on social organization patterns. Specifically, they reported that smaller chimpanzee communities formed larger, more stable mixed-sex subgroups in comparison to large chimpanzee communities [24,27]. Thus, wild chimpanzees are able to navigate complex and frequently changing social networks and need to put effort into establishing and maintaining strategic relationships.

However, in captivity, chimpanzees do not have control or choice regarding with whom or with how many others they share their enclosure, i.e., form a group. Furthermore, changes in group composition, i.e., chimpanzees joining or leaving a captive population, are much less frequent as well as tend to be of permanent nature in comparison to the flexible fission–fusion system of their wild conspecifics. Here, the group size tends to be limited mostly by the enclosure size and the occurrence of aggressive events, while changes to the group composition are often based on breeding programs, the resulting offspring or chimpanzees passing away [39]. Although issues such as food availability and other pressures related to survival may not apply to captive populations, these chimpanzees lack the possibility to split or switch between subgroups in order to stay close or avoid specific group members.

Several accredited organizations, including the Association of Zoos and Aquariums (AZA), suggest the optimal group size lies between 7 and 9 individuals, recommending mixed-sex groups, with several males and females of different ages as well as ideally more females than males [40]. However, it is more likely to find captive groups to be smaller, lacking age diversity and consisting of more males than females [41]. Generally, there tends to be a surplus of males in captivity, and thus housing organizations end up forming all-male groups, which may lead to excessive agonistic events [24]. Due to the lack of females, all-male groups are unable to exhibit socio-sexual behaviors, depriving them of a natural behavioral motivator. Although chimpanzees tend to be very flexible and capable of adjusting and adapting to new social situations [42,43,44], factors such as a stagnant group size, lacking the presence of the opposite sex and unbalanced group composition restrain their natural potential and impact their behavior [45,46,47]. Several studies demonstrated that social housing produces a positive impact on chimpanzee health and behavior, as living in a functional social group provides constantly changing stimulation and challenges the animals socially and cognitively [47,48], making it one of the most important and most effective environmental enrichments [49]. Sanctuaries, working only with one or few primate species, aim to establish bigger, more complex social groups, allowing a certain amount of choice and control based on environmental and care management strategies promoting these aspects [9,50]. Yet more research is needed in order to predict the most suitable group compositions in captivity.

The sociability of chimpanzees is reflected in the exhibition of behaviors such as grooming [51], socio-sexual interactions [52,53], maternal–filial relationships [54,55], social play [56,57] and hierarchical relationships [15]. Infant chimpanzees spend their first two to five years close to their mothers [14,58,59], developing their social skills by interacting with their mother and other members of their group [14], learning and acquiring species-typical behaviors [60], which includes acquiring social capacities to navigate social complex networks. However, many chimpanzees with a past as pets, entertainers or laboratory animals did not experience such an upbringing [61]. During the second half of the 20th century, thousands of infant chimpanzees were captured from the wild [62,63], typically resulting in early maternal loss [64,65], prolonged social isolation [66] and the lack of social partners during their first years of life [67,68]. Experiencing the traumatic change from living in the wild to life in captivity, the deplorable conditions during transport and the lack of maternal and peer relations immensely increases the probability of developmental problems [69,70,71]. Several studies demonstrated that early maternal deprivation and social isolation in wild-caught chimpanzees influence their emotional, psychological and physiological development and future rehabilitation capacities [72,73], as well as increasing their susceptibility to stress, in comparison to chimpanzees raised by their mothers [74]. On the other hand, chimpanzees born in captivity possess a higher recovery potential than those caught in the wild [45]. Thus, such adverse experiences and living conditions during infancy negatively impact the chimpanzee’s potential to live and function in a social group [75]. Yet, several studies have shown that chimpanzees with a past of adverse living conditions and traumatic experiences can still recover to some degree and be socially integrated over time [76,77]. Furthermore, the excessive exhibition of undesired behaviors, such as abnormal behaviors, high frequencies of self-directed behaviors and inactivity, can be reduced by providing species with adequate living conditions and professional care [45]. Several studies looking into the social capacities of former pet, entertainer and laboratory chimpanzees found that chimpanzees’ grooming activity and distribution were significantly impacted by early life adversities such as their origin, the onset of maternal deprivation and predominant social housing condition during infancy [78,79]. They reported wild-caught, early deprived and/or chimpanzees housed predominantly without conspecifics during infancy to exhibit much lower frequencies of grooming and distribute their grooming attention unequally, i.e., selective grooming partner choice. Furthermore, Crailsheim et al. (2020) found former pet and entertainment chimpanzees to adapt their grooming patterns after alterations to the group composition occurred, irrespective of their early life experience. Thus, wild-caught chimpanzees are more likely to have lost or never acquired natural social abilities. Lacking basic social skills and/or social experience is likely to produce a lasting social impairment, which remains detectable even once they are provided with a more adequate social environment. Hence, it is likely that wild-caught chimpanzees might be less capable of taking advantage of improved social housing conditions in comparison to captive-born chimpanzees, i.e., maintain low levels of social grooming activity, regardless of the housing organization’s efforts to improve aspects regarding the group composition.

Allogrooming is one of the most important social tools used by chimpanzees [14] serving several functions, such as hygiene [80], establishing and maintaining social relationships [81] and stress regulation [64]. High levels of self-directed grooming are considered undesirable [82]; on the other hand, high levels of social grooming in captivity can be considered an indicator of positive welfare and success regarding social integration [83]. In the wild, group size varies according to season, being larger during the dry season compared to the wet season [16,22,25,31]. Chimpanzees tend to increase and expand their grooming activity with increasing group size, although it may also vary due to ecological pressures, such as the habitat and/or seasonality [46]. Major seasonal differences in temperature, rainfall and thus food availability, i.e., seasonality, produce behavioral adaptions in many primate species, including chimpanzees [16,17,24,25,30,31,38,84,85]. In order to adapt to food scarcity, chimpanzees modify their feeding activity [86,87] and adjust their grouping patterns and group size [24,84,88] by forming smaller groups and reducing the time spent on social interactions [84,89]. In contrast, in environments where resources are less limited, no differences in group size between seasons have been observed [90]. During times when food resources are abundant, travel time is reduced, while time spent on resting and social interactions is likely to increase [84,91]. Furthermore, resting has been documented to be positively correlated with temperature, with individuals remaining more inactive at higher temperatures [85].

In the wild, only females reaching sexual maturity emigrate from one community to another [18,28] and form bonds with adult males in order to settle in [92,93,94]. Males remain in the natal group and perform territorial functions [14]. Chimpanzees are highly territorial [14] and may engage in intense and occasionally lethal aggression towards individuals from neighboring communities [32,95,96,97]. Regarding social introductions of adult chimpanzees in captivity, this partially explains the higher success rate of female–female or female–male introductions compared to those between males [98,99]. Exchange between captive populations, i.e., the transfer of individuals from one group to another, is not limited to sexually mature females, since rescue centers take in chimpanzees regardless of their age or sex and zoos enact breeding programs. A successful integration implies not only the absence of excessive aggression [100], but also the establishment of relationships and the presence of affiliative interactions with other group members [83,101,102]. Several studies report successful social integrations without the occurrence of serious injury [50,103,104], while others report encountering major difficulties [98,99]. During the first few days after the integration of new individuals, agonistic social behaviors are likely to increase [103,104], although there are strategies to limit aggressive escalations. According to Schel et al. [104], a well-designed and equipped enclosure may allow individuals to avoid or escape aggressive situations.

Aside from sex and age, another factor that might influence the success rate of an integration is the life history, as integrations between individuals who experienced similar living conditions during their early life are more likely to succeed [98]. Successful introductions, in the long term, result in a significant reduction in agonistic behaviors between the new individuals and the original group and a significant increase in affiliative relationships [104]. In order to draw these conclusions, long-term observation projects are necessary as short-term observations only provide information regarding a small timeframe that could still reflect only the adaptation to the new social setting.

The alteration of a group composition is not limited to the introduction of new individuals but also includes animals leaving a group in order to join another population or due to the loss of life.

Analyzing social dynamics has become increasingly popular over the last few decades, and social network analysis (SNA) is a very useful tool allowing us to better understand complex social systems [105,106]. It allows us to describe, quantify and statistically compare the social relationships of individuals within a group and how changes over time impact the social dynamics on an individual and group level [107,108,109]. This tool can be used to examine how social behavior patterns change over time [110,111] and thus can support efforts regarding animal wellbeing and care management [112].

For this study, we analyzed seven years of observational data on a group of former pet and entertainment chimpanzees which went through two major group alterations during that time. Originally being an all-male group, alterations to the group composition resulted in changes regarding both the group size and sex ratio. Rather than looking at the direct impact during the alteration event itself, we were interested in the long-term effect and how the individual and social behavior patterns would be affected. Furthermore, by using observational data over such an extended time frame, we were able to take seasonality into account, which is expected to have a significant impact on the chimpanzees’ behavior.

Our primary aim was to demonstrate the magnitude of factors potentially impacting the lives of captive-housed chimpanzees by considering aspects regarding their (adverse) past and depending on an extended data collection. Thus, we strive to demonstrate that evaluations regarding a chimpanzee’s behavior, wellbeing and/or housing conditions could easily be misinterpreted if they are based on short-time observations or do not consider the animal’s background information. Specifically, we investigated the impact of the following factors on chimpanzee behavior: (1) Considering that adequate social housing is one of the most effective tools for increasing the wellbeing of captive-housed chimpanzees, we expected group alterations to impact the chimpanzees’ activity levels, the frequency of undesired behaviors and their grooming patterns. Specifically, we expected an increase in group members as well as the introduction of females, i.e., the possibility to exhibit socio-sexual behaviors, to a former all-male group to result in lower frequencies of undesired behaviors, higher activity levels and potentially more social activity. (2) We strive to highlight the importance of longitudinal data collection and seasonal effects by demonstrating different levels of activity and social interactions depending on the climatic situation. Specifically, we expected chimpanzees to increase levels of activity, both on an individual level and on a social level, during the colder season. (3) Taking into account previous findings regarding the long-term impact of chimpanzees’ adverse past, we expected wild-caught chimpanzees to present a certain behavioral impairment, i.e., to be less active and less sociable and exhibit higher frequencies of undesired behaviors.

## 2. Materials and Methods

### 2.1. Sample Population and Study Site

The present study was conducted at the primate rescue center Fundació MONA, a center dedicated to the rescue, rehabilitation and life-long care of primates coming from illegal and/or species-inadequate living conditions, member of the European Alliance of Rescue Centers and Sanctuaries (EARS), in Riudellots de la Selva (Girona, Spain). During the time frame of this study, MONA housed a total of 13 former pet and entertainment chimpanzees living in two social groups named Mutamba and Bilinga. The study population of the current study consists of seven individuals belonging to Mutamba, five males and two females (Table 1), with the two females being the individuals that were transferred from the Bilinga to the here-observed Mutamba group. This results in a total of 20 possible dyads, including 10 male–male dyads, 9 female–male dyads and 1 female–female dyad.

During seven years of data collection, two major group alterations occurred in the Mutamba group due to the integration of two females and the death of two males. These alterations produced three different social environments, with all other aspects of their living conditions remaining the same: (1) all-male group consisting of five males; (2) mixed-sex group consisting of five males and two females; (3) mixed-sex group consisting of three males and two females (Table 2). Each social setting/phase was observed for over a year including data on both cold and warm seasons.

Observations were conducted from an observation tower while the chimpanzees were residing in or had access to the outdoor enclosure throughout the day. Said outdoor enclosure contains natural substrate (Mediterranean vegetation), measures 2420 m^2^ in size and is surrounded by a steel mesh and electric fence. Chimpanzees were granted access to the outdoors from 10 AM to 7 PM (varies throughout the year depending on the climate conditions) but would spend the nights in the indoor areas. The outdoor enclosures are equipped with several climbing structures, such as wooden platforms, towers and other structures, as well as climbing ropes, enrichment devices and hammocks. A wall of vegetation, mainly bamboo (*Phyllosta chysaurea*), has been planted around the perimeter in order to shelter and isolate the animals from undesired noise and stimulation from the surroundings. Throughout the study, chimpanzees had ad libitum access to water dispensers and were fed 4–5 times a day based on a diet plan consisting of vegetables, fruits, legumes, dried fruits, seeds and protein-rich items. For more detailed information regarding the chimpanzees’ living conditions and habitat design, see [113,114].

### 2.2. Data Sampling

Data on the chimpanzees’ behavior were recorded over 70 months from June 2016 to March 2022, resulting in a total of 1400 h of observation data and 220,087 recorded scans. Observers conducted instantaneous scan sampling [115], simultaneously recording data on all chimpanzees present in the observed enclosure, based on 2 min intervals during 20 min sessions. Sessions were randomly distributed throughout the days of the week and evenly distributed between the morning and afternoon hours while animals had access to the outdoor enclosure. An ethogram (Appendix A) established by Fundació MONA was used, consisting of 17 behavioral items, 16 sub-behaviors and information regarding the social partner and directionality of social interactions. Observers were only allowed to collect data once they finished a training period and successfully passed a three-step interobserver reliability test. The first step included data collection over a minimum of two weeks; these data were checked and then deleted. In the second step, observers had to pass a methodology test; in the third step, they had to pass a video test, identifying animals and behaviors with an agreement of ≥ 85% with the research coordinator of the center.

### 2.3. Statistical Analysis

In order to evaluate the chimpanzees’ behavior depending on seasonality, aspects regarding their social environment and past, the behavioral database was organized into six separate time frames. Thus, all variables of interest have been calculated per individual for each phase and seasonality.

#### 2.3.1. Social Network Analysis (SNA)

To analyze the social grooming data, we calculated the percentage of scans engaged in grooming by counting the number of scans where individual A groomed individual B, and by dividing this number by the total number of scans where both individuals were actually present in the same enclosure and then multiplied the proportion by 100. Thus, we took into account the time dyads actually had access to each other during the observations, as individuals occasionally would be absent for short amounts of time due to veterinary issues, veterinary training or other animal care management reasons. We created our networks in UCINET 6.758 [116] using NetDraw 2.179 [117] for visual graph representation. The weighted network graphs consist of nodes representing the chimpanzees and directed edges representing the percentage of scans each chimpanzee spent grooming its group members. Data on social grooming were used to calculate the following two network measures which have been previously described by Kasper and Voelkl [118] and used by Kalcher-Sommersguter et al. [65]. For a detailed explanation regarding the calculation and interpretation of these network measures, you may consult the previously mentioned sources.

Vertex strength centrality (VSC): This index is a measure describing the standardized strength of an individual’s grooming activity. More precisely, it reflects the mean percentage of scans an individual spent grooming another individual in his/her group, while taking the group size into account.

Deviation from edge weight disparity (DEWD): The edge weight disparity is a measure describing how evenly a chimpanzee is distributing his/her grooming among all group members. By calculating the deviation from this edge weight disparity, we can compare the grooming distribution between groups of different group sizes.

#### 2.3.2. Indicators of Behavioral Wellbeing

To analyze the animals’ activity levels, a general activity index (GAI) was calculated per individual for each phase and seasonality. Based on the behavioral records, we classified the observed behaviors as “active”, including agonistic interactions, affiliative interactions, behaviors related to food acquisition and feeding, behaviors directed at humans, locomotion, manipulation of enrichment or the environment, socio-sexual interactions and solitary play, or “inactive”, including resting related behaviors, stationary vigilance behavior, abnormal behaviors and self-directed behaviors. Behaviors were assigned to these labels based on the physical requirements, indicating either an active or an inactive state. Observed abnormal behaviors consisted of 85% excessive self-grooming, 8% repetitive self-scratching and some occurrences of coprophagy and self-poke. To calculate this index, the following formula was used: GAI = (Active − Inactive)/(Active + Inactive). The obtained values range between −1 and 1, with higher values indicating elevated levels of activity and 0 indicating an equal occurrence of activity and inactivity levels. Similar indices have proved to be useful in other behavioral studies based on activity budgets [119].

To analyze the occurrence of undesired behaviors (i.e., abnormal and self-directed behaviors), we divided the number of scans in which an individual performed undesired behaviors by the total amount of recorded scans per individual. We are aware that “self-directed” behaviors are not necessarily categorized as “undesired” yet decided to include this behavioral category here because low frequencies of self-directed behaviors would only impact this variable slightly and would not be considered a behavioral problem. However, high frequencies of self-directed behaviors, such as frequent yawning, scratching and continuous attention to oneself rather than others and the environment, are likely to reflect elevated levels of stress, anxiety and/or a lack of stimulation [120,121] and as such fit the criteria of “undesired”. These calculations were performed for all group members, for each phase and seasonality.

#### 2.3.3. Linear Mixed Models (LMMs)

We ran four separate LMMs to assess the impact of the fixed factors group size (5 vs. 7), sex ratio (all-male vs. mixed-sex), seasonality (cold vs. warm), sex (M vs. F) and origin (wild-caught vs. captive-born). As the dependent variables, we used the previously explained variables: vertex strength centrality, deviation from edge weight disparity, general activity index and undesired behaviors. To account for repeated measures on the same individuals, we included the chimpanzee ID as a random factor. To control the normal distribution of the residuals, we visually checked the QQ plots, which revealed no violations of the assumptions of our LMMs. All models were run using the “lme4” package [122] in R 3.5.0 [123]. The multicollinearity between all fixed factors was tested by calculating the variance inflation factor (VIF), using the “car” package [124]. The VIFs calculated for the five fixed factors used in our models ranged between 1.00 and 1.61, indicating that our fixed factors were not correlated. To confirm the quality of our LMMs, we used the “anova” function comparing the full model (including all fixed factors) with the null model (without fixed factors) to ensure that the fixed factors had a significant effect on the model outcome [125]. The significance of the fixed factors was then explored by “anova” analysis, and post hoc tests were performed using the “emmeans” and “emtrends” functions with Bonferroni adjustments. Additionally, we used the package “effsize” to calculate the effect size using Cohen’s d; the results are presented in the Appendix A.

## 3. Results

We ran four LMMs to investigate the potential impact of group size, sex ratio, seasonality, sex and origin.

### 3.1. Graphical Representation of Grooming Networks

The graphical representations of the weighted social grooming networks of all six observation time periods based on the group composition and seasonality are shown in Figure 1. Nodes represent group members, with square nodes being male and circle nodes being female chimpanzees. The node color represents which individual joined or left the group, with black being individuals who were present throughout all phases, red being the individuals who joined (in phase 2) and blue being the individuals who left (in phase 3).

The graphical representations permit us to quickly spot networks missing potential interaction dyads. Specifically, phase 1, during the warmer season, is the only network missing two interaction dyads (TON-JUA and TON-BON), making this the least complex network in our study sample. When comparing networks belonging to the colder and the warmer season conditions, we see the lines, representing the grooming activity between individuals, are slightly thicker, indicating higher levels of grooming during the colder season. Regarding the complexity of the networks, due to the increase in potential grooming partners, phase 2 (N = 7) is clearly more complex, i.e., socially stimulating, in comparison to phases 1 and 3 (N = 5). Furthermore, potential interaction dyads missing in the previous phase (phase 1, warm) are present in phase 2, in both season conditions.

### 3.2. Effects on Grooming Pattern

The full model regarding deviation from edge weight disparity was not improved by including the fixed factors, and thus none of the predictors had any impact on the distribution of the chimpanzees’ grooming attention (Appendix A).

Grooming activity (VSC) was significantly influenced by the seasonality, the chimpanzees’ sex and origin (Appendix A). With respect to seasonality, we found individuals to exhibit a significantly higher grooming activity during the cold season (mean temperature: 14.45 ± 3.44 °C) compared to the warm season (mean temperature: 24.78 ± 5.04 °C), F(1,27) = 7.89, *p* < 0.01, d = 0.44 (Appendix A, Figure 2). Regarding sex, we found that females groomed their group members significantly more frequently than males, F(1,8) = 8.80, *p* < 0.05, d = 0.88, (Appendix A, Figure 2). With respect to origin, we found captive-born chimpanzees to have a significantly higher grooming activity than wild-caught chimpanzees, F(1,7) = 9.56, *p* < 0.05, d = 1.02 (Appendix A, Figure 2). However, fixed factors regarding the group composition failed to show any significant effect on the grooming activity (Appendix A).

### 3.3. Effects on General Activity Levels

The model regarding the general activity index was significantly influenced by group size, sex ratio, seasonality and the chimpanzees’ origin (Appendix A). With respect to group size, the post hoc test shows that activity was significantly higher when the group was larger (seven individuals) compared to a smaller group size of five individuals, F(1,28) = 14.06, *p* < 0.001, d = 0.02 (Appendix A, Figure 3). However, this significance has to be interpreted with caution, as the effect size is very low. Furthermore, we found that in the all-male setting, chimpanzees exhibited significantly higher activity levels compared to the mixed-sex group setting, F(1,28) = 25.33, *p* < 0.001, d = 0.83 (Appendix A, Figure 3). Regarding the chimpanzees’ origin, we found captive-born chimpanzees to have a significantly higher activity level compared to wild-caught chimpanzees, F(1,8) = 27.33, *p* < 0.001, d = 1.98 (Appendix A, Figure 3). Seasonality significantly affected the chimpanzees’ activity levels, with higher levels during the cold season (mean temperature: 14.45 ± 3.44 °C) compared to the warm season (mean temperature: 24.78 ± 5.04 °C), F(1,27) = 25.47, *p* < 0.001, d = 0.73 (Appendix A, Figure 3). However, the factor sex failed to show any significant effect on the chimpanzees’ activity levels (Appendix A).

### 3.4. Effects on the Exhibition of Undesired Behaviors

The model based on undesired behaviors was significantly influenced by the group´s sex ratio and the chimpanzees’ origin (Appendix A). In relation to the group´s sex ratio, the post hoc test shows a significantly higher occurrence of undesired behaviors in the all-male group compared to the mixed-sex group setting, F(1,27) = 7.74, *p* < 0.01, d = −1.87 (Appendix A, Figure 4). Regarding origin, we found wild-caught chimpanzees to present a significantly higher exhibition of undesired behaviors than captive-born chimpanzees F(1,7) = 12.57, *p* < 0.01, d = 0.84 (Appendix A, Figure 4). However, the factors sex, seasonality and group size failed to show any significant effect on the occurrence of undesired behaviors (Appendix A).

## 4. Discussion

By conducting long-term observations on a group of captive chimpanzees going through two major group modifications within the last seven years, we could demonstrate that group composition, seasonality and the chimpanzees’ biographic background produce an impact on the chimpanzees’ behavior. Specifically, low-temperature periods were marked by higher grooming and general activity levels. A bigger group size seemed to have a stimulating effect, as chimpanzees exhibited higher activity levels. While being housed as an all-male group, males exhibited higher frequencies of undesired behaviors. Chimpanzees labeled as wild-caught generally exhibited less grooming, lower levels of activity and higher levels of undesired behaviors.

Based on official recommendations regarding captive social housing [40] and by taking chimpanzees’ sociability in the wild into account [126], we should assume that in phase 2, a bigger mixed-sex group would be the preferable social environment for chimpanzees to thrive and show indications of improved wellbeing, i.e., higher levels of activity and social grooming and less exhibition of undesired behaviors. Furthermore, by looking at how these chimpanzees modify their relationships after group alterations occur, we can gauge their adaptability to changes in their social environment, i.e., group alterations.

Chimpanzees are highly flexible and capable of adjusting to new social situations [43,44,127], but modifications in group size and sex ratio do have a strong impact on their behavior [23,45,128]. In captivity, due to management and maintenance needs, actions involving changes in group structure and composition are necessary at times. These changes tend to be long-lasting and depend exclusively on care management decisions rather than on resource availability or individual preferences. In this study, we were able to make use of long-term behavioral observations of a captive chimpanzee population experiencing changes to its social environment, while maintaining all other aspects of their captive living conditions. We found that chimpanzees did modify their behavior, i.e., are affected by the new social group composition, yet no changes in the amount of grooming given or its distribution to group members were detected. This might indicate that considering their constant living conditions, including captive care and environment, the amount of grooming recorded in phase 1 (i.e., smaller all-male group) already represents the maximum amount of grooming an individual is willing to engage in. This amount might be enough investment to maintain and/or establish new relationships with members of the group. Furthermore, Crailsheim et al. [78] found that grooming distribution, i.e., the choice of how to distribute one’s grooming attention to group members, was affected by group alterations, but would be more evenly distributed in the long term, during stable social settings. Thus, not finding any significant impact on the grooming distribution was to be expected in this study, as each phase was documented for a minimum of one year. Although the amount of grooming observed in the all-male phase did not differ much compared to other phases, some males were not observed grooming each other at all. Hence, they were not taking advantage of all social partners at that time even though the group size was considered small. However, we did detect changes in their general activity levels and the occurrence of undesired behaviors, and these changes only partially concur with our expectations. Undesired behaviors were indeed more frequently recorded when no females were part of the group and the group consisted only of five individuals, which is estimated to be the least ideal social group composition. On the other hand, general activity levels, which were expected to be positively correlated with a bigger group size and the presence of females, were only higher when considering the group size but were not influenced by the mixed-sex setting. Furthermore, since the effect size of this variable was very low, the significant result regarding the group size has to be treated with caution. The increase in group size is likely to produce a stimulating effect, resulting in the observed increase in general activity levels [14,50,98,99]. Nevertheless, the presence of females allowed males to exhibit socio-sexual behaviors such as frequent genital inspection of the other sex, which they were unable to do as an all-male group. The recuperation of sexual motivations and behaviors in their ethogram has to be considered as positive for the males, yet it did not produce the expected result when comparing all-male and mixed-sex activity levels. Considering that both group size and sex ratio significantly affected the general activity levels in our activity model, it would be interesting to analyze the interaction between these two factors. However, due to the small sample size, this is not recommended as the resulting model would not be sufficiently robust.

It is important to the authors to always bear in mind how unique each individual chimpanzee actually is in terms of his/her personality and past experiences, thus showing differences in preferences and capacities. Here we included biogeographical information regarding their sex and origin (wild-caught vs. captive-born) in order to partially address behavioral differences between individuals. Boesch and Boesch-Achermann, working with wild populations [24,45], state that males are more gregarious and show higher grooming rates than females, suggesting this to be strongly influenced by dispersal patterns and habitat quality [129]. In environments where resource competition is less pressing and females do not emigrate, such as in captivity, several studies found females to exceed males in their grooming activity, showing their social potential [42]. Our data further support these findings in captive populations, as we found females to be more active groomers compared to males. Yet, the individuals’ sex did not influence any other behavioral outcome within our study.

Chimpanzees [14] require a complex physical and social environment with sufficient stimulation to ensure their wellbeing [130]. The first years of life are crucial for their future development, and bonds with the mother and other conspecifics are essential to provide adequate learning opportunities and allow practicing social skills [69,70,71,131,132]. Deprivation of these relationships during early life may lead to problems in chimpanzees’ emotional, physical and social development, thus affecting their capacities to function in social groups later in life [71,75]. It might even lead to mood and anxiety disorders [133] and elevated levels of stress [74]. All chimpanzees of this study sample have experienced a variety of adverse living conditions, lacking species-adequate care before arriving at the sanctuary. While we could not validate the exact onset of maternal deprivation, extent of human exposure or details regarding their commercial or private use, information regarding their origin has been confirmed. Several studies suggest that, in captivity, wild-caught chimpanzees spend less time on social activities, such as actively grooming others, compared to captive-born chimpanzees [45,78,79]. One possible explanation might be that wild-caught chimpanzees experienced additional traumatic experiences, such as witnessing the death of their mother and/or group members, suffering extreme conditions during capture and transportation, and eventually a dramatic change in their living conditions from wild to captivity [134,135,136]. By extreme conditions, we refer to the temporary circumstances during the active trafficking of the wild-caught chimpanzees, such as being held in a tiny, poorly ventilated transport box without any thermoregulation. These traumatic events can be expected to be reflected in their social grooming activity and behavior. As expected here as well, wild-caught chimpanzees were less active groomers compared to the captive-born individuals, although no differences in their distribution of grooming attention have been found. This impairment, however, is not limited to their social activity but is also detectable in their individual behavior budget. Several studies suggest that traumatic experiences during early life stages in chimpanzees are likely to result in lower activity levels and facilitate the emergence of abnormal behaviors in chimpanzees [137,138,139]. On the contrary, chimpanzees born in captivity tend to be more active [45] and spend less time on self-directed behaviors [45,139]. Independent of the alterations to the group composition, we also found wild-caught chimpanzees to be significantly less active, exhibiting higher frequencies of undesired behaviors and engaging less in grooming activities compared to the captive-born chimpanzee in our populations. Thus, we provide further evidence of past traumatic experiences, and here origin, creating a certain impairment regarding the chimpanzees’ behavior, although these results must be considered with the limitation of the small sample size of seven individuals. However, in view of our findings, we suggest it is important to consider the background of the animals (adverse past living conditions) during the formation of new groups or when introducing new members into a group. Specifically, we recommend expecting wild-caught chimpanzees to be less active and less sociable in general. On the contrary, captive-born chimpanzees may have the potential to instigate social interactions and activity within their group, thus possibly invigorating less active and less sociable group members such as wild-caught chimpanzees.

Furthermore, we were able to highlight the importance of long-term observations as these allowed us to control the impact of seasonality and be certain of grasping the impact of the social environment (long term) rather than only the impact of the group alteration event itself, i.e., integration or exit of individuals (short term). In the wild, chimpanzees have been documented to modify their behavior according to season [84], generally due to resource availability [16,17,24,25,30,31,38] but also due to differences in environmental temperature [85]. Several studies suggest that when resources decrease, group size decreases, and individuals engage in more solitary behavior as they need to spend more time on food acquisition [84,89]. In captivity, resource availability does not vary, as feeding by caregivers remains constant over time. On the other hand, depending on the geographical location, the temperature might vary significantly throughout the year. The MONA sanctuary, being located in the north of Spain, is influenced by the Mediterranean climate, showing relatively big differences between the colder and warmer months (cold season: 14.5 ± 3.4 °C; warm season: 24.8 ± 5.0 °C). Several studies suggest resting to be positively correlated with temperature, with individuals remaining more inactive at higher temperatures [85,140]. In addition, inactivity in intense heat situations might be associated with energy conservation [85,141]. Temperature variations may also affect the social relationships of primates; it has been observed that Japanese macaques (*Macaca fuscata*) spend more time in proximity while experiencing very low temperatures [142]. In line with these previous studies, our results also suggest that seasonality and temperature have a significant impact on the chimpanzees’ behavior in captivity. Here, both the general activity levels and the social grooming activity were significantly higher during colder months compared to warmer months. Several reasons might explain these behavioral patterns, such as (1) searching for increased physical contact and social proximity in order to maintain or raise body temperature during colder days, (2) energy conservation during very hot days and (3) limiting activities during warmer months to shaded areas which tend to be relatively small in captive habitats. This finding highlights the importance of taking the regional climatic tendencies into consideration when comparing chimpanzee behaviors between captive populations that are located in different countries (e.g., Spain vs. England) or between observation periods regarding the same population but conducted in different seasons.

We are very aware of the limiting factor of our sample size in terms of individuals, yet we believe that the extensive observation time and amount of behavioral data used in this study allow us to produce meaningful results and specifically permit us to highlight the importance of longitudinal observations. Many research projects are limited by tight deadlines and the pressure to produce results in the shortest amount of time possible. However, we believe it is of utmost importance to extend data collection in order to objectively evaluate the animals’ behaviors while considering factors that cannot be taken into account in short observation periods. Here, specifically, we could show that seasonality had a major impact. If this study would have been conducted based on short observation periods, not taking seasonality into account, we would have run the risk of misinterpreting the impact of seasonality as being produced by other factors such as the group composition. Thus, we demonstrated that long-term observations allow us to include more potentially influencing factors and make our results more likely to objectively reflect the animals’ behaviors and state of wellbeing. We highly recommend that primate housing organizations conduct behavioral observations to monitor animal behavior and wellbeing. Additionally, we suggest setting up extended observation periods equally distributed throughout the year, rather than arranging infrequent, short but concentrated observation schedules.

## 5. Conclusions

We were able to demonstrate that group alterations, climatic conditions and biographic information have an impact on chimpanzees’ behavior to some degree. Group alterations have an impact on general activity levels and the occurrence of undesirable behaviors, but not on the time spent in social grooming. The impact of early life adversities on social grooming and general activity levels and the exhibition of undesired behaviors could be detected even years after an individual’s arrival at the sanctuary. Seasonality affected grooming activity and general activity levels, which were higher during low-temperature periods.

We highly suggest that comparisons between different groups or evaluations over time should consider the above-mentioned predictors and be interpreted with caution. We are well aware of the relatively small sample size used in this study, and although we made an effort to compensate for the lack of individuals by increasing the detail and time frame of the data collection, we do not claim that these factors are the only factors influencing the chimpanzees’ behavior.

Regarding the observed group compositions, we suggest the larger mixed-sex group to be the most preferable option. As none of the grooming indices varied, we base this conclusion on the general behavior patterns. A bigger group size might have had a stimulating impact, increasing the chimpanzees’ general activity levels. In the absence of females, males exhibited higher frequencies of undesired behavior. On the contrary, in the mixed-sex setting, chimpanzees were able to express socio-sexual behaviors that males previously could not exhibit.

We wish to emphasize once more the importance of using long-term observations when assessing primate behavior in captivity, as there are bound to be fluctuations and many potentially influential factors that can only be controlled by such long-term datasets.

## Figures and Tables

**Figure 1 animals-13-00424-f001:**
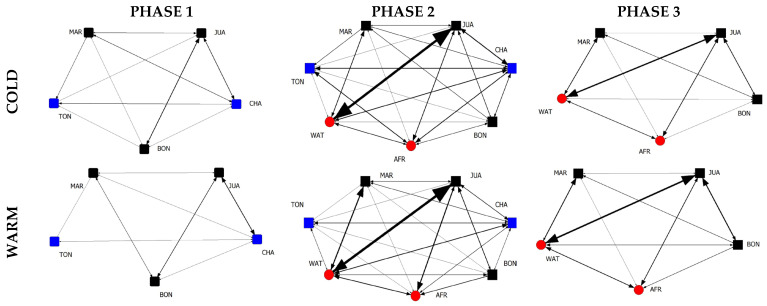
Social grooming networks of all 6 observation time periods based on the group composition (phase 1 vs. phase 2 vs. phase 3) and seasonality (cold vs. warm).

**Figure 2 animals-13-00424-f002:**
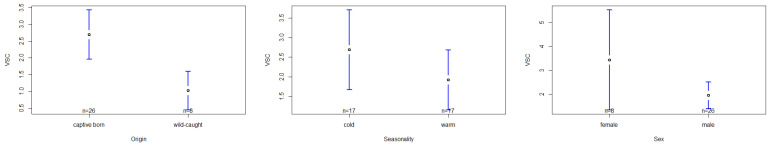
Confidence interval plots of the VSC and all significant fixed factors (seasonality, sex and origin).

**Figure 3 animals-13-00424-f003:**
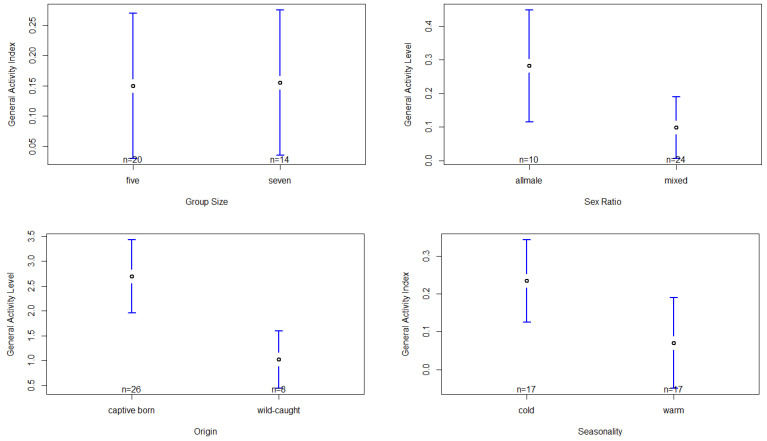
Confidence interval plots of the general activity levels and all significant fixed factors (size, sex ratio, origin, seasonality).

**Figure 4 animals-13-00424-f004:**
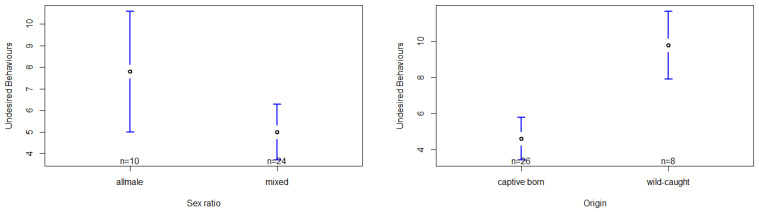
Confidence interval plots of undesired behaviors and the two significant fixed factors (sex ratio, origin).

**Table 1 animals-13-00424-t001:** Individuals’ characteristics and background information.

Name	ID	Sex	Estimated Year of Birth	Origin	Year of Introduction to the Group
Africa	AFR	F	2000	wild-caught	2017
Bongo	BON	M	2000	captive-born	2009
Charly *	CHA	M	1989	captive-born	2001
Juanito	JUA	M	2003	captive-born	2003
Marco	MAR	M	1984	captive-born	2001
Toni *	TON	M	1983	wild-caught	2001
Waty	WAT	F	2002	captive-born	2017

Abbreviations: M = male, F = female. * Died in 2020.

**Table 2 animals-13-00424-t002:** Chronology of the different observations time periods.

Observation Phase	Observation Time Frame	Group Size	Group Composition	Individuals	Group Alteration
Phase 1	2016–2017	N = 5	All-male	BON, CHA, JUA, MAR, TON	Original all-male group
Phase 2	2017–2020	N = 7	Mixed-sex	AFRI, BON, CHA, JUA, MAR, TON, WAT	Integration of two females
Phase 3	2021–2022	N = 5	Mixed-sex	AFR, BON, JUA, MAR, WAT	Two males died

## Data Availability

Data is contained within the Appendix A.

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
