# Peer review of "Long-Term Assessment of Captive Chimpanzees: Influence of Social Group Composition, Seasonality and Biographic Background"

_animals, 2023, doi:10.3390/ani13030424_

Round 1
Reviewer 1 Report
Congratulations on an interesting and informative project. The results will be of interest to a wide audience and will hopefully inspire a change of approach to chimpanzee care. The results are particularly interesting due to the extended data collection period and the consideration of wild vs captive-bred.
The methods and analyses are sound and well summarized/presented. The Discussion is comprehensive and the conclusions merited.
The only comments I have relate to written style and English:
Line 18: mayor should read ‘major’
Line 26: Habitat’s should read habitat, or ‘the habitat’s
Line 29: define LMMs
Line 70: should read ‘….ranging in size between…..’
Line 87: community should read ‘communities’
Line 114: should be chimpanzee
Line 116: should read ‘….challenges an animal’s….’
Line 153: should read ‘….grooming, as well as distributing…..’ (remove the and)
Line 170: should read ‘spent’
Line 189 and Line 458: mayor should read ‘major’
Line 242: in is repeated
Table 1 ‘captivity born’ should read ‘captive born’
Line 264: does not read well – please review
Line 282: multifocal implies several individuals were observed at once, while scan implies the entire group was sampled. Can you clarify?
Line 291: should read ‘these data were checked…..’
Line 298: database – no gap
Line 304: were should read ‘where’
Line 306: no comma after time
Line 318 and Line 324: replace his with ‘their’
Line 319: should read ‘it was calculated by…’
Line 320: should read ‘…..si by the group number – 1, (N-1)…’
Line 330: should read ‘this was obtained by calculating….’
Line 349: replace upper-case S with lower case s
Line 350: Not sure what this sentence means – the physical requirements?
Line 375: add a ‘included as a random factor…..’
Line 378: explain LTR, explain VIF
Line 394: move the explanation of VIF to line 378
General comment on analysis – can you add in and later comment on effect sizes in addition to significance?
Line 466: should read ‘official’
Line 470: should read ‘less exhibition of undesirable behaviours…’ so need to add and s to les
Line 479: chimpanzee’s should read ‘chimpanzee’
Line 486: sentence should end with ‘in’
Line 507: should read ‘in the long term’
Line 520: recommendable should read ‘recommended’
Line 546: I think ‘grooming given’ means the amount they groom another – can this be clarified please?
Line 549: extreme condition – can you explain? Might mean ‘extreme trauma’
Line 607: bee should read ‘been’
Line 614: wording would be improved by stating ‘We demonstrated that…..’
Line 702: Wrangham reference incomplete
Line 710-711: No need for upper case font
Reference list needs to be formatted consistently (journal titles for example)
Line 776: McGrew, WC is repeated
I very much enjoyed reading this manuscript. Thank you.
Author Response
Author answer: Thank you very much for your feedback and your very motivating response/opinion regarding the research topic itself. We personally are always worried that studies regarding such basic information on captive chimpanzees, might not be glamourous enough for some journals, yet strongly believe that this type of research is essential to further improve our capacities to efficiently care for these animals in captivity.
Thank you very much for taking the time to not only evaluate the science behind this study, but also spot our grammar and spelling mistakes. We are very sorry we didn´t spot the spelling mistakes ourselves and created additional work for you.
We did our best to correct all detected mistakes and improve the sections mentioned by the reviewer and are very grateful for the help to further improve this manuscript.
The only comments I have relate to written style and English:
Line 18: mayor should read ‘major’ (corrected)
Line 26: Habitat’s should read habitat, or ‘the habitat’s (corrected)
Line 29: define LMMs (corrected)
Line 70: should read ‘….ranging in size between…..’ (corrected)
Line 87: community should read ‘communities’ (corrected)
Line 114: should be chimpanzee (corrected)
Line 116: should read ‘….challenges an animal’s….’ (corrected)
Line 153: should read ‘….grooming, as well as distributing…..’ (remove the and) (corrected)
Line 170: should read ‘spent’ (corrected)
Line 189 and Line 458: mayor should read ‘major’ (corrected)
Line 242: in is repeated (corrected)
Table 1 ‘captivity born’ should read ‘captive born’ (corrected)
Line 264: does not read well – please review (thank you, we now refrased the enclosure explanation)
Line 282: multifocal implies several individuals were observed at once, while scan implies the entire group was sampled. Can you clarify? (thank you, we slightly changed the wording making sure the readers understand that the whole group was observed simultaneously using scan sampling based on 2-minute intervals)
Line 291: should read ‘these data were checked…..’ (corrected)
Line 298: database – no gap (corrected)
Line 304: were should read ‘where’ (corrected)
Line 306: no comma after time (corrected)
Line 318 and Line 324: replace his with ‘their’ (corrected)
Line 319: should read ‘it was calculated by…’ (corrected)
Line 320: should read ‘…..si by the group number – 1, (N-1)…’ (corrected)
Line 330: should read ‘this was obtained by calculating….’ (corrected)
Line 349: replace upper-case S with lower case s (corrected)
Line 350: Not sure what this sentence means – the physical requirements? (thank you, we now moved this phrase a bit further up, to make sure readers understand we refer to the labels “active” and “inactive” and not to the abnormal behaviors. We also modified the phrase a bit, explaining that the physical requirements refer to the behaviors listed from our ethogram, which were than either categorized as an active or inactivity belonging item)
Line 375: add a ‘included as a random factor…..’ (corrected)
Line 378: explain LTR, explain VIF (thank you, we now tried to make this technical section a bit more reader friendly. The LTR test referred to “testing/ensuring the quality” of our Full models. As it is a very technical term not often used in papers, we now rephrased this part. We also added a better explanation regarding the VIF: The VIF stands for variance inflation factor which serves as an indicator to avoid certain combinations of fixed factors in case a high collinearity (correlation) between fixed factors were to be detected.)
Line 394: move the explanation of VIF to line 378 (corrected)
General comment on analysis – can you add in and later comment on effect sizes in addition to significance? (We now calculated effect size as well, adding this information to the post-hoc analysis tables in the supplementary material.)
Line 466: should read ‘official’ (corrected)
Line 470: should read ‘less exhibition of undesirable behaviours…’ so need to add and s to les (corrected)
Line 479: chimpanzee’s should read ‘chimpanzee’ (corrected)
Line 486: sentence should end with ‘in’ (corrected)
Line 507: should read ‘in the long term’ (corrected)
Line 520: recommendable should read ‘recommended’ (corrected)
Line 546: I think ‘grooming given’ means the amount they groom another – can this be clarified please? (thank you, you are correct, we now changed this to make it clearer.)
Line 549: extreme condition – can you explain? Might mean ‘extreme trauma’ (thank you, we now added more explanation: With extreme conditions, we refer to the temporary circumstances during active trafficking, such as typically being held in a tiny box, poorly ventilated without any thermoregulations.)
Line 607: bee should read ‘been’ (corrected)
Line 614: wording would be improved by stating ‘We demonstrated that…..’ (corrected)
Line 702: Wrangham reference incomplete (corrected)
Line 710-711: No need for upper case font (corrected)
Reference list needs to be formatted consistently (journal titles for example) (corrected)
Line 776: McGrew, WC is repeated (corrected)
I very much enjoyed reading this manuscript. Thank you.
Reviewer 2 Report
Dear Authors. I have reviewed your manuscript which I find it very interesting and necessary to gain more knowledge to improve the wellbeing of highly intelligent animals in captivity. As you mention you have only three cases (phases) to compare and this has to be considered when conclusions are drawn. However, data on behavior of mammals in zoo´s are scars and hopefully your data will inspire others to look at optimal social constellations for chimpanzee and other primates. In general try to be more precise when you describe your results, some examples are given below, but try to do this in general. Also, I suggest that you focus on the challenges to keep chimpanzees in zoos - and the differences between behaviors in wild-caught and captive born chimpanzees. What will you recommend to do in order to improve socialization of the two groups with different origin. Do you recommend that chimpanzees are caught in the wild to live in zoos?
Line 25: “These changes depend mostly on habitat’s resource availability and individual social preferences” – add in the wild or in nature.
Line 33: “occurrence of undesired behaviors” please mention some examples of both desired and undesired activity.
Line34: “Seasonality did influence the activity level and time spent grooming” in short in what way?
Line 35: “Moreover, it was the individuals’ origin but not changes in group composition that affected social grooming”? Try to be more precise - e.g. Confidence interval plots showed that general activity and grooming activity was lower in captive born chimpanzees, while on the contrary undesired behaviour was higher in wild caught chimpanzees.
Line 37: “Alterations of their social environment may be considered both stimulating and stressful, yet long term evaluations are necessary to predict and detect potential benefits and/or issues”. This is rather out in the blue- try to use your results to conclude something that zoo´s may use to make the captive life of both captive born and wild born chimpanzees more comfortable.
Line 45-49: These lines are irrelevant for your focus; you never discuss the similarities with humans again. Your data provide knowledge about the behavior of chimpanzees and your main focus should be that individuals are so influenced by their origin that it will influence their behaviour despite social context.
Line 49-155: Try to focus your introduction on challenges with chimpanzees caught in the wild and born in captivity- pets or chimpanzees that may have lost natural social abilities. You have all the elements, but could you make it more chronologically.
Line 220-233. Your aim is not clear – it covers 13 lines. Try to make your aims more precise and short. What do you want your study to contribute with in order to help zoos to improve conditions for chimpanzees. What are your hypotheses your hypotheses maybe list them or list with numbers.
Line 233: agonistic interactions are they always an expression of wellbeing?
Line 298: Try to describe these sociograms in words- not only show the graphs. How do you interpret them. The social grooming networks is seems more complex in phase 2.
Line 45. The discussion should be shortend and base your discussion more chronological on your main results. It can be improved and try to conclude with some advises how to improve social and desired activities and avoiding undesired activities in the two groups-captive and wild born chimpanzees.
Author Response
Author answer: Thank you very much for your feedback and your very motivating response/opinion regarding the research topic itself. We personally are always worried that studies regarding such basic information on captive chimpanzees, might not be glamourous enough for some journals, yet strongly believe that this type of research is essential to further improve our capacities to efficiently care for these animals in captivity.
Thank you very much for taking the time to not only evaluate the science behind this study, but also help to improve the readers experience by suggesting modifications in how to order and express our findings.
We did our best to modify all indicated sections mentioned by the reviewer and are very grateful for the help to further improve this manuscript.
Line 25: “These changes depend mostly on habitat’s resource availability and individual social preferences” – add in the wild or in nature. (thank you, corrected using the previous phrase)
Line 33: “occurrence of undesired behaviors” please mention some examples of both desired and undesired activity. (we added information here)
Line34: “Seasonality did influence the activity level and time spent grooming” in short in what way? (we explained the impact in more detail)
Line 35: “Moreover, it was the individuals’ origin but not changes in group composition that affected social grooming”? Try to be more precise - e.g. Confidence interval plots showed that general activity and grooming activity was lower in captive born chimpanzees, while on the contrary undesired behaviour was higher in wild caught chimpanzees. (we explained the impact in more detail)
Line 37: “Alterations of their social environment may be considered both stimulating and stressful, yet long term evaluations are necessary to predict and detect potential benefits and/or issues”. This is rather out in the blue- try to use your results to conclude something that zoo´s may use to make the captive life of both captive born and wild born chimpanzees more comfortable. (we changed this phrase)
(Comments regarding the abstract were taken into account as much as possible, though we had to be minimalistic, due to the limited word count of the abstract into account. In order to add the required information, we had to delete and reduce other parts, but strived to heed the suggestion of the reviewer as much as possible.)
Line 45-49: These lines are irrelevant for your focus; you never discuss the similarities with humans again. Your data provide knowledge about the behavior of chimpanzees and your main focus should be that individuals are so influenced by their origin that it will influence their behaviour despite social context. (we slightly reduced this part to reduce the emphasis, but maintained parts as its purpose is to explain that “what makes these animals especially interesting to most humans, i.e. their intelligence, awareness and complex needs, also makes them species extremely difficult to care for and to provide a welfare promoting environment for.)
Line 49-155: Try to focus your introduction on challenges with chimpanzees caught in the wild and born in captivity- pets or chimpanzees that may have lost natural social abilities. You have all the elements, but could you make it more chronologically. (while we would love to place more emphasis on the variable of their background as we did in several other publications previously, we try not to overstate this topic due to the low number of individuals of our group in this study. That being said we added a few more sentences in the end of the mentioned section)
Line 220-233. Your aim is not clear – it covers 13 lines. Try to make your aims more precise and short. What do you want your study to contribute with in order to help zoos to improve conditions for chimpanzees. What are your hypotheses your hypotheses maybe list them or list with numbers. (We tried to be more precise towards our aims and added numbering to structure the three different hypotheses. We also add an explanation of what we hoped to find regarding the seasonality. Further explanations regarding the befits were now added in the discussion/conclusion)
Line 233: agonistic interactions are they always an expression of wellbeing? (agonistic interactions are part of the chimpanzee’s way of interacting and communicating. Thus, agonistic interaction itself should never be evaluated as something negative. However excessive or extreme agonistic interaction on an intra-group level can be considered dangerous and are likely to reduce the individuals ‘wellbeing)
Line 298: Try to describe these sociograms in words- not only show the graphs. How do you interpret them. The social grooming networks is seems more complex in phase 2. (yes, this was indeed missing, we are very sorry to not have done this from the start)
Line 45. The discussion should be shortend and base your discussion more chronological on your main results. It can be improved and try to conclude with some advises how to improve social and desired activities and avoiding undesired activities in the two groups-captive and wild born chimpanzees. (we shortened some paragraphs and added some recommendations based on our findings)